# Acetaminophen-Induced Hepatotoxicity in Obesity and Nonalcoholic Fatty Liver Disease: A Critical Review

**Karima Begriche, Clémence Penhoat, Pénélope Bernabeu-Gentey, Julie Massart**  **and Bernard Fromenty \***

INSERM, Univ Rennes, INRAE, Institut NUMECAN (Nutrition Metabolisms and Cancer) UMR_A 1341, UMR_S 1241, F-35000 Rennes, France
\* Correspondence: bernard.fromenty@inserm.fr; Tel.: +33-2-23-23-30-44

**Abstract:** The epidemic of obesity, type 2 diabetes and nonalcoholic liver disease (NAFLD) favors drug consumption, which augments the risk of adverse events including liver injury. For more than 30 years, a series of experimental and clinical investigations reported or suggested that the common pain reliever acetaminophen (APAP) could be more hepatotoxic in obesity and related metabolic diseases, at least after an overdose. Nonetheless, several investigations did not reproduce these data. This discrepancy might come from the extent of obesity and steatosis, accumulation of specific lipid species, mitochondrial dysfunction and diabetes-related parameters such as ketonemia and hyperglycemia. Among these factors, some of them seem pivotal for the induction of cytochrome P450 2E1 (CYP2E1), which favors the conversion of APAP to the toxic metabolite N-acetyl-*p*-benzoquinone imine (NAPQI). In contrast, other factors might explain why obesity and NAFLD are not always associated with more frequent or more severe APAP-induced acute hepatotoxicity, such as increased volume of distribution in the body, higher hepatic glucuronidation and reduced CYP3A4 activity. Accordingly, the occurrence and outcome of APAP-induced liver injury in an obese individual with NAFLD would depend on a delicate balance between metabolic factors that augment the generation of NAPQI and others that can mitigate hepatotoxicity.

**Keywords:** acetaminophen; drug-induced liver injury; obesity; nonalcoholic fatty liver disease; steatosis; nonalcoholic steatohepatitis; diabetes; cytochrome P450 2E1; fatty acids; mitochondria



## 1. Introduction

The epidemic of obesity is associated with a steady rise in drug consumption in order to treat several associated diseases such as type 2 diabetes mellitus (T2DM), hypertension, atherosclerosis, dyslipidemia and osteoarthritis [1,2]. In addition, numerous drugs are currently being developed in order to specifically treat nonalcoholic fatty liver disease (NAFLD), which is frequently associated with obesity and overweight [3,4]. This implies increased polypharmacy among obese patients, which can augment the risk of adverse events including drug-induced liver injury (DILI) [5,6]. In line with this, recent investigations reported a higher frequency of DILI in patients with NAFLD [7,8]. More specifically, the common pain reliever acetaminophen (APAP) is one of the identified drugs that could be more hepatotoxic in obesity and NAFLD, at least after an overdose [9]. The present article reviews the clinical and experimental investigations published on APAP-induced liver injury in the context of these metabolic diseases and also discusses the possible reasons that might explain why some studies are discrepant from others. Because our previous review on this matter was published in 2014 [9], many recent investigations are now discussed in this updated review.

## 2. APAP Hepatotoxicity

### 2.1. General Overview

APAP, also referred to as paracetamol, is one of the most widely prescribed drugs for the management of pain and hyperthermia. The current maximum recommended

dosage of APAP is 4 g/day in adults even though the Food and Drug Administration (FDA) advises doses below 3.25 g/day for chronic use [10]. Although therapeutic doses of APAP can induce hepatic cytolysis in some patients [11,12], most cases of severe APAP-induced acute liver injury occur after accidental or intentional overdoses [13,14]. Actually, APAP is deemed to have a narrow therapeutic margin since as little as 7.5 g/day might be hazardous [15]. Currently, administration of N-acetylcysteine (NAC) is the only approved therapy to treat APAP overdose-induced liver injury in patients [10,14]. The rationale of NAC administration is to restore hepatic levels of glutathione (GSH), a major endogenous antioxidant limiting the noxious effects of the APAP toxic metabolite N-acetyl-*p*-benzoquinone imine (NAPQI) (Figure 1) [16,17]. Notably, repeated or long-term intake of APAP at therapeutic doses can occasionally cause acute hepatic cytolysis of different severities [11,12] but also chronic liver injury such as granulomatous hepatitis and cirrhosis [18–20].

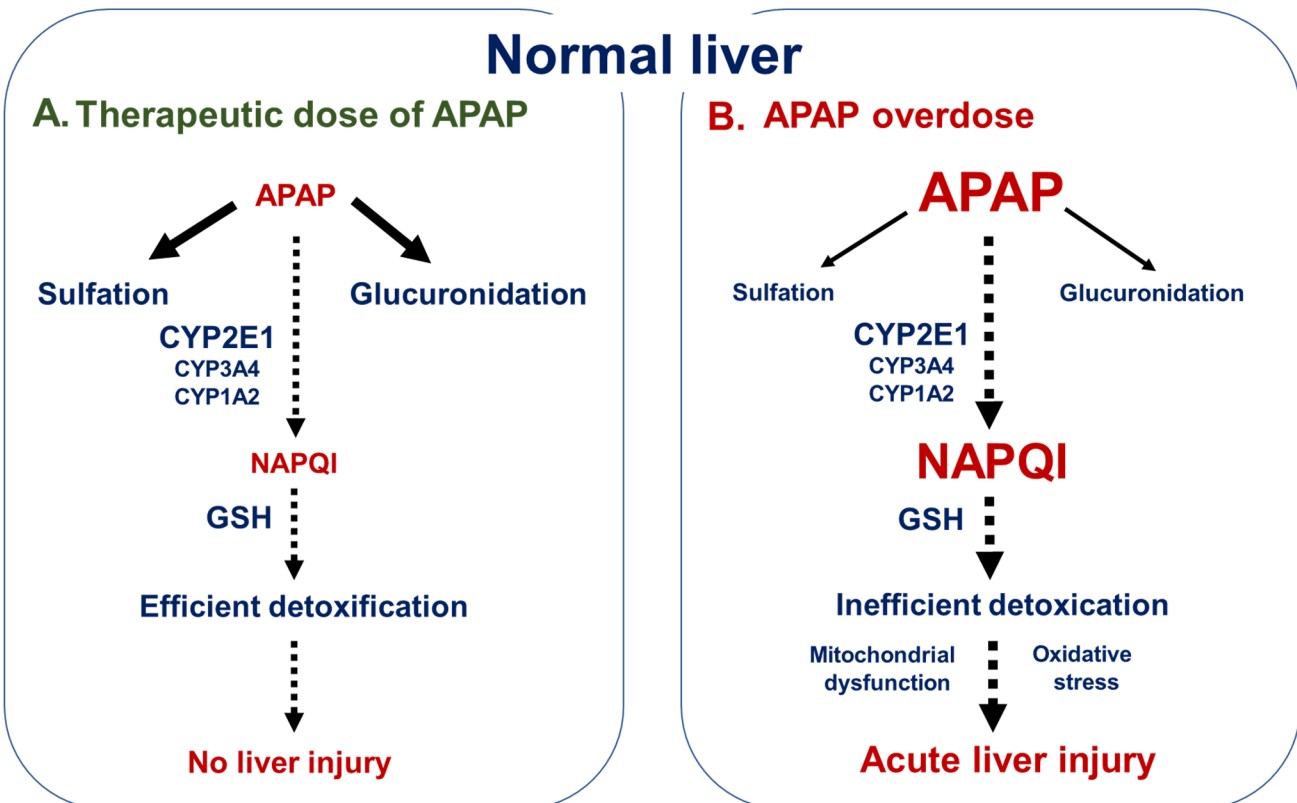

**Figure 1.** Biotransformation and toxicity of APAP in normal liver. (**A**). For therapeutic dose, APAP is mainly detoxified through sulfation and glucuronidation, while a small proportion is metabolized to N-acetyl-*p*-benzoquinone imine (NAPQI) via the cytochrome P450 2E1 (CYP2E1) and to a lesser extent CYP3A4 and CYP1A2. In normal liver, the low amounts of NAPQI are efficiently detoxified by glutathione (GSH), a major antioxidant molecule present in different cellular compartments including mitochondria. (**B**). After APAP overdoses, the sulfation and glucuronidation pathways are overwhelmed and more APAP undergoes CYP-dependent oxidation to NAPQI. However, GSH concentrations in hepatocytes are not sufficient to allow the efficient detoxification of NAPQI, which then induces major mitochondrial dysfunction, oxidative stress and acute liver injury. More information is provided in the text.

A key player in APAP liver injury is cytochrome P450 2E1 (CYP2E1), an enzyme that catalyzes the oxidation of APAP to NAPQI (Figure 1) [10,21,22]. Indeed, NAPQI is a highly reactive metabolite inducing severe mitochondrial dysfunction, overproduction of reactive oxygen species (ROS), and c-jun N-terminal kinase (JNK) activation, eventually leading to ATP depletion and massive hepatocellular necrosis [17,22,23]. Importantly, mitochondrial CYP2E1 could play a major role in APAP-induced cytotoxicity [24,25]. Finally, CYP3A4

(referred to as CYP3A2 in rats and CYP3A11 in mice) and CYP1A2 might also play a role in the conversion of APAP to NAPQI (Figure 1), although to a lesser extent than CYP2E1 in normal physiological conditions [26,27].

### 2.2. Predisposing Factors

Except for APAP ingested dose, APAP-induced hepatotoxicity could be favored by different factors such as chronic alcohol abuse, severe or chronic liver diseases, prolonged fasting and malnutrition, older age, and some comedications such as antituberculosis and antiepileptic drugs [12,16,28,29]. Importantly, increased activity of hepatic CYP2E1 (and possibly other CYPs) seems to be a common mechanism whereby chronic alcohol abuse, prolonged fasting and some comedications favor APAP-induced liver injury [16,30]. As discussed in this review, obesity, NAFLD and both types 1 and 2 diabetes could also predispose to APAP liver injury, at least in part, due to higher hepatic CYP2E1 activity [9,10,13]. Finally, the risk of APAP hepatotoxicity could be modulated by polymorphisms in different genes [16], such as *UGT1A* encoding UDP-glucuronosyltransferase (UGT) 1A, which plays a pivotal role in APAP glucuronidation and detoxification (Figure 1) [28,31].

## 3. APAP Hepatotoxicity in NAFLD

### 3.1. Main Features of NAFLD

Because of the epidemic of obesity and T2DM, NAFLD is now the most frequent chronic liver disease worldwide with a global prevalence of 25% [32]. NAFLD comprises a large spectrum of histologic changes including simple fatty liver, nonalcoholic steatohepatitis (NASH), advanced fibrosis and cirrhosis [33], which can evolve into hepatocellular carcinoma (HCC) [34]. It is estimated that simple fatty liver progresses to NASH in about 10 to 20% of the patients [35]. NASH itself is defined by the presence of steatosis (mostly macrovacuolar), some necrosis and apoptosis, hepatocellular ballooning and lobular inflammation [33]. Of note, the presence of microvesicular steatosis has been associated with histological markers of NASH severity [36]. Although the mechanisms of progression of fatty liver to NASH in some patients are not fully understood, mitochondrial dysfunction, oxidative stress and lipid peroxidation are deemed to play a primary role in the occurrence of cell death and inflammation [37–39].

### 3.2. Clinical Investigations on Acute APAP Hepatotoxicity in Obesity and NAFLD

There is some clinical evidence that obesity and NAFLD can predispose to APAP hepatotoxicity in the setting of APAP overdose (Table 1). Two large retrospective studies reported that APAP-induced acute liver injury was more frequent in NAFLD patients [40,41]. In these studies, patients with pre-existing NAFLD hospitalized for APAP overdose had a four- to sevenfold higher prevalence of acute liver injury as compared to those without NAFLD [40,41]. In another study, APAP-induced acute liver injury was more frequent in overweight or obese patients, but NAFLD presence was not investigated [42]. Obesity might also favor APAP hepatotoxicity when this analgesic and antipyretic drug is taken at therapeutic doses. Indeed, mild to moderate hepatic cytolysis, as evidenced by increased plasma transaminases (ALT and AST), was reported in some morbidly obese patients but not in nonobese individuals after receiving 4–5 g of intravenous APAP [43].

In contrast to these studies, the occurrence of APAP-induced acute liver injury was reported to be similar, or even lower, in obese patients compared to nonobese individuals (Table 1) [44,45]. However, one of these studies showed that obese patients had significantly poorer clinical outcomes after acute liver failure [44]. The discrepancies between the aforementioned studies might arise from several factors including the degree of obesity, the existence of NASH and advanced fibrosis and the presence of insulin resistance and T2DM. Indeed, these factors could alter APAP absorption, distribution, metabolism and excretion (ADME) but also basal antioxidant defenses and mitochondrial function, as discussed in Section 4.

**Table 1.** Summary of the clinical studies (ordered by increasing year) carried out on APAP-induced acute liver injury in obesity and NAFLD.

| Authors, Year [References] | Design of the Study | Presence of NAFLD | Hepatic CYP2E1 Activity | APAP Overdose | APAP-Induced Acute Liver Injury |
|---|---|---|---|---|---|
| Rutherford et al., 2006 [44] | Prospective | Not reported in this study [1] | Not reported in this study | Yes | Lower incidence (but poorer outcomes) in obese patients |
| Nguyen et al., 2008 [40] | Retrospective | Yes | Not reported in this study | Yes | Higher prevalence in patients with NAFLD |
| Myers and Shaheen, 2009 [41] | Retrospective | Yes | Not reported in this study | Yes | Higher prevalence in patients with NAFLD |
| Radosevich et al., 2016 [45] | Retrospective | Not reported in this study [1] | Not reported in this study | Yes | Equal prevalence between obese and nonobese patients |
| Van Rongen et al., 2016 [43] | Prospective | Not reported in this study [1] | Increased | No (4 to 5 g) | Increased plasma ALT and AST in morbidly obese patients but not in nonobese individuals |
| Chomchai and Chomchai, 2018 [42] | Retrospective | Not reported in this study [1] | Not reported in this study | Yes | Higher prevalence in overweight and obese patients |

[1] There is now ample evidence that obesity is strongly associated with NAFLD (reviewed in [32,35]).

### 3.3. Rodent Studies on Acute APAP Hepatotoxicity in Obesity and NAFLD

APAP-induced acute hepatotoxicity has also been investigated in different rodent models of obesity and NAFLD (Table 2). However, while several investigations reported greater APAP hepatotoxicity in obese rodents [9,46–53], others showed no difference or even lower APAP-induced liver injury compared to lean rodents [9,46,54–57]. In addition to some factors mentioned in the previous section, discrepancies between these experimental investigations might be due to differences in the rodent model (rats vs. mice), the origin of obesity (genetic vs. diet-induced) and the composition of the hypercaloric diet, as discussed in Section 4.

In some aforementioned investigations, APAP not only caused more severe hepatic cytolysis in obese mice (as evidenced by increased ALT and AST) but also worsened liver fat accumulation through a mechanism that might involve inhibition of autophagy and exacerbation of oxidative stress [52,53]. Interestingly, aggravation of steatosis was also observed in ob/ob mice acutely intoxicated with APAP although this was not associated with higher plasma transaminases and more severe hepatic necrosis [49]. In NAFLD, distinct mechanisms might thus be involved in APAP-induced hepatic cytolysis and worsening of steatosis, respectively.

**Table 2.** Summary of the rodent studies (ordered by increasing year) carried out on APAP-induced hepatotoxicity in obesity and NAFLD.

| Authors, Year [References] | Rodent Models of Obesity and NAFLD | Presence of NAFLD | Hepatic CYP2E1 Activity | Dose of APAP | APAP-Induced Hepatotoxicity |
|---|---|---|---|---|---|
| Corcoran and Wong, 1987 [47] | Male Sprague–Dawley rats fed a high-fat diet for 24 weeks | Not reported in this study [1] | Not reported in this study | 710 mg/kg (i.p.) | Higher hepatotoxicity after 48 h, compared to rats fed a standard diet |
| Blouin et al., 1987 [58] | Male obese Zucker fa/fa rats | Not reported in this study [2] | Not reported in this study [2] | 1300 mg (p.o.) | Similar hepatotoxicity after 48 h, compared to lean rats |

**Table 2.** *Cont.*

| Authors, Year [References] | Rodent Models of Obesity and NAFLD | Presence of NAFLD | Hepatic CYP2E1 Activity | Dose of APAP | APAP-Induced Hepatotoxicity |
|---|---|---|---|---|---|
| Tuntaterdtum et al., 1993 [54]. | Male obese Zucker fa/fa rats | Not reported in this study [2] | Not reported in this study [2] | 3000 mg/kg (p.o) | Lower hepatotoxicity after 48 h, compared to lean rats |
| Ito et al., 2006 [55] | Male C57Bl/6 mice fed a Western-style diet for 16 weeks | Yes | Not reported in this study | 300 mg/kg (p.o.) | Lower hepatotoxicity after 6 h, compared to mice fed a standard diet |
| | Male ob/ob mice | Not reported in this study [3] | Not reported in this study [3] | 300 mg/kg (p.o.) | Lower hepatotoxicity after 6 h, compared to wild-type mice |
| Donthamsetty et al., 2008 [59] | Male Swiss Webster mice fed a MCD diet for 1 month [4] | Yes | Unchanged | 360 mg/kg (i.p.) | Higher hepatotoxicity from 6 to 48 h after overdose, compared to mice fed a standard diet |
| Kon et al., 2010 [48] | Male KK-A[y] mice | Yes | Not reported in this study [5] | 300 or 600 mg/kg (p.o.) | Higher hepatotoxicity after 6 h, compared to wild-type mice |
| Kucera et al., 2012 [50] | Male Sprague-Dawley rats fed a high-fat diet for 6 weeks | Yes | Not reported in this study | 1 g/kg (p.o) | Higher hepatotoxicity after 24 and 48 h, compared to rats fed a standard diet |
| Aubert et al., 2012 [49] | Female db/db mice | Yes | Increased | 500 mg/kg (p.o.) | Higher hepatotoxicity after 8 h, compared to wild-type mice |
| | Female ob/ob mice | Yes | Unchanged | 500 mg/kg (p.o.) | Similar hepatotoxicity after 8 h, compared to wild-type mice |
| Kim et al., 2017 [56] | Male C57Bl/6 mice fed a fast food diet for 14 weeks | Yes | Not reported in this study (but higher CYP2E1 protein levels) | 200 mg/kg (i.p.) | Lower hepatotoxicity compared to wild-type mice (timing not specified) |
| Piccinin et al., 2019 [51] | Male FVB/N mice fed a high-fat diet for 1 month | Yes | Not reported in this study | 300 mg/kg (i.p.) | Higher hepatotoxicity after 6 h, compared to wild-type mice |
| Shi et al., 2019 [52] | Male C57Bl/6 mice fed a high-fat diet for 8 weeks | Yes | Not reported in this study | 50, 100 or 200 mg/kg (p.o.) | Significant hepatotoxicity after 24 h but no comparison with wild-type mice |
| Wang et al., 2021 [53] | Male C57Bl/6J mice fed a high-fat diet for 8 weeks | Yes | Not reported in this study | 100 mg/kg (p.o.) | Significant hepatotoxicity after 24 h but no comparison with wild-type mice |
| Ghallab et al., 2021 [57] | Male C57Bl/6N mice fed a Western diet for 48 to 50 weeks | Yes | Not reported in this study (but lower CYP2E1 immunostaining) | 300 mg/kg (i.p.) | Lower hepatotoxicity compared to wild-type mice (timing not specified) |

[1] Numerous investigations in rodents including rats showed that long-term feeding of high-fat diets consistently induces NAFLD (reviewed in [60–62]). [2] Other studies showed that male obese and insulin resistant Zucker fa/fa rats present moderate fatty liver [63,64] but reduced CYP2E1 activity [63,65]. [3] Other investigations showed that male obese and diabetic ob/ob mice present major fatty liver [66,67], with unchanged [49] or reduced [68] CYP2E1 activity. [4] Methionine and choline-deficient (MCD) diet is known to induce NASH, which is however associated with reduced body weight and blood glycemia [9,60]. [5] Previous studies showed that hepatic CYP2E1 mRNA expression [69] and activity [70] are unchanged in KK-A[y] mice.

*3.4. In Vitro Studies on Acute APAP Hepatotoxicity in Models of Fatty Acid Exposure and NAFLD*

Several in vitro studies investigated APAP acute cytotoxicity in different models of fatty acid exposure and NAFLD. Two studies were carried out in hepatocytes isolated from rats fed different types of lipids. The first study reported that liver slices from rats fed a diet rich in butter (which mainly contains saturated fatty acids) were significantly more sensitive to APAP cytotoxicity than those from rats fed a diet enriched in polyunsaturated fatty acids (PUFAs) [71]. Unfortunately, lipid accumulation was not evaluated in this study nor were included liver slices from rats fed a standard diet. Nevertheless, this study suggests that exposure to long-chain saturated fatty acids could be more detrimental than to polyunsaturated linoleic acid (C18:2) and arachidonic acid (C20:4) [71]. In the second study, steatotic primary hepatocytes isolated from rats fed a diet enriched in corn oil (which mainly contains PUFAs) were more sensitive to APAP cytotoxicity than those from rats fed a standard diet [72]. The role of n-3 PUFAs (also referred to as ω-3 PUFAs) in APAP hepatotoxicity is discussed in Section 4.2.5.

Two other studies were performed in hepatocyte cell lines incubated with different fatty acids. In the first study, carried out in L02 liver cells, the investigations showed that a 24 h exposure to the monounsaturated oleic acid (C18:1) exacerbated APAP cytotoxicity whereas different medium chain fatty acids did not cause this effect [73]. Unfortunately, this study did not determine whether these different fatty acids induced steatosis in L02 liver cells. Other investigations performed in differentiated HepaRG cells incubated 7 days with stearic acid (C18:0) or oleic acid (which both induced steatosis) showed that only stearate supplementation induced greater APAP-induced cytotoxicity, which was blunted by the CYP2E1 inhibitor chlormethiazole [74]. The apparent discrepancy between these two studies could be due to the cell lines and the duration of oleic acid exposure. Nonetheless, these in vitro investigations clearly indicate that exposure to some fatty acids could favor APAP hepatotoxicity. Although this might be due to their propensity to induce CYP2E1, other possible mechanisms cannot be excluded, as briefly discussed in Section 4.1.3.

Finally, in vitro investigations also reported that APAP worsened lipid deposition in steatotic L02 cells [52,53], thus confirming in vivo results in diet-induced and genetically obese mice [49,52,53]. However, steatosis in L02 cells was induced by cotreating the cells with oleic acid and ethanol, which does not reflect pure NAFLD. Nevertheless, these investigations suggest that acute APAP could aggravate steatosis through a direct effect on hepatocytes and not via extrahepatic pathways such as fat mobilization from adipose tissue [75,76].

*3.5. Investigations on Chronic APAP Hepatotoxicity in Obesity and NAFLD*

Repeated or chronic intake of therapeutic doses of APAP can sporadically cause different types of liver injury, as previously mentioned [18–20]. Unfortunately, there are no clinical studies investigating the occurrence of repeated or chronic APAP hepatotoxicity in obesity and NAFLD. In rodents, a 13-week treatment with APAP was less hepatotoxic in leptin receptor-deficient obese (fa/fa) Zucker rats than in lean rats [77]. According to the authors, this might be explained by lower hepatic CYP2E1 expression in obese Zucker rats [77]. This is in line with previous studies showing downregulation of hepatic CYP2E1 in obese Zucker rats [63,68] and lower acute APAP hepatotoxicity in obese Zucker rats compared with their lean littermates [54]. The role of the adipokine leptin in CYP2E1 expression is briefly discussed in Section 4.2.2. In another study, a 35-day treatment with APAP caused more severe hepatic cytolysis in spontaneously diabetic torii (SDT) rats as compared to nondiabetic rats [78]. While SDT rats are not obese, this study did not investigate fatty liver [78]. APAP hepatotoxicity in type 1 diabetes is discussed in Section 6.

**4. Factors Modulating APAP Hepatotoxicity in Obesity and NAFLD**

From the abovementioned studies carried out in humans and rodents, it appears that obesity and NAFLD do not always increase the risk or the severity of APAP-induced liver

injury. Hence, while several factors would favor APAP hepatotoxicity in these metabolic diseases, others might limit APAP toxicity.

*4.1. Factors That Could Favor APAP Hepatotoxicity in Obesity and NAFLD*

4.1.1. CYP2E1 Induction

Hepatic CYP2E1 induction could be a major mechanism associated with greater APAP hepatotoxicity observed in most clinical and experimental studies (Figure 2A), although other explanations can be considered as discussed below. Indeed, higher CYP2E1 activity is expected to cause an overproduction of NAPQI and deeper GSH depletion, thus leading to more severe mitochondrial dysfunction and oxidative stress [9,13,79]. In line with this hypothesis, the study by van Rongen et al. reported higher CYP2E1 activity in morbidly obese patients, which was associated with mild to moderate hepatic cytolysis after administration of 4–5 g of APAP [43]. However, CYP2E1 activity was not determined in the other investigations reporting a higher risk of APAP-induced liver injury in patients with obesity and NAFLD [40–42]. Experimentally, investigations in ob/ob and db/db obese mice showed that APAP hepatotoxicity correlated with hepatic CYP2E1 activity but not with liver fat accumulation [49]. Unfortunately, most other rodent studies showing higher APAP hepatotoxicity in obese animals did not investigate CYP2E1 expression, or activity [47,48,50–53].

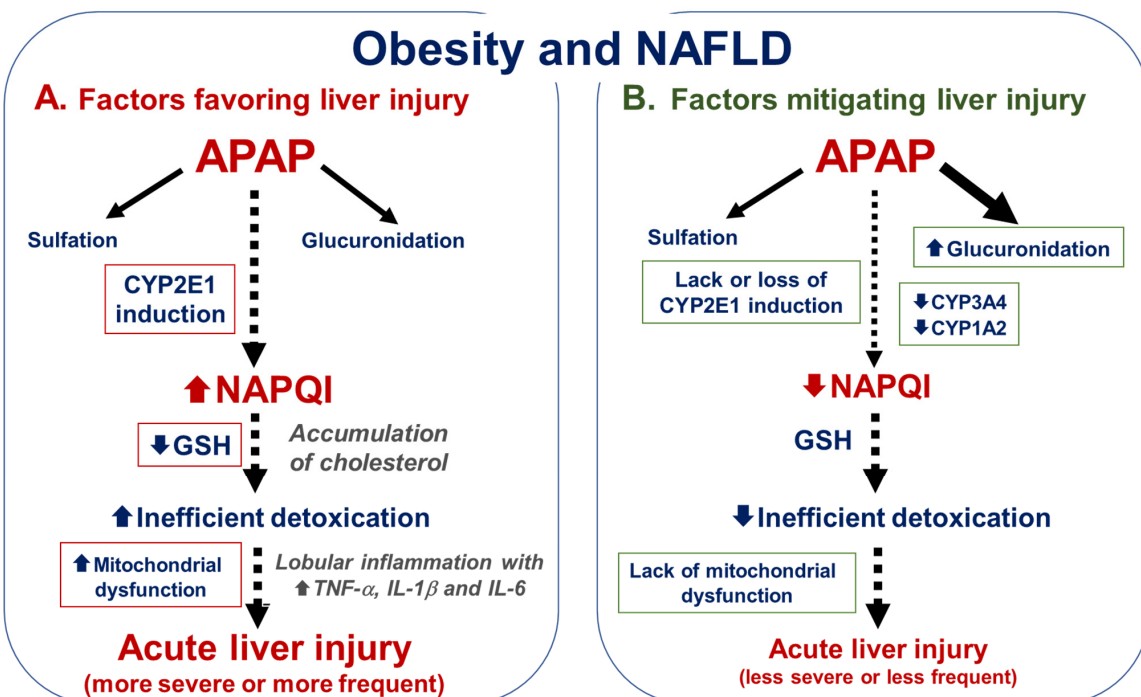

**Figure 2.** Hepatotoxicity of APAP in obesity and NAFLD. (**A**). Different factors in obesity and NAFLD could favor liver injury induced by APAP overdose, for instance by increasing cytochrome P450 2E1 (CYP2E1) activity, reducing basal concentrations of glutathione (GSH) and promoting preexisting mitochondrial dysfunction. In addition, the accumulation of cholesterol could sensitize the liver to APAP-induced hepatotoxicity by favoring mitochondrial GSH depletion. Lobular inflammation might also favor APAP hepatotoxicity via several cytokines such as tumor necrosis factor-α (TNF-α), interleukin-1β (IL-1β) and IL-6. (**B**). On the contrary, some factors in obesity and NAFLD could mitigate APAP-induced liver injury, for instance by increasing APAP glucuronidation and reducing CYP3A4 and CYP1A2 activity. Moreover, CYP2E1 induction could be absent or lost in some metabolic and pathological conditions. The absence of preexisting mitochondrial dysfunction in some patients might also mitigate APAP-induced hepatotoxicity. Consequently, obese people with one or several of these mitigating factors might not have a higher risk of severe APAP-induced liver injury. More information is provided in the text.

In a study carried out in mice fed a fast food diet enriched in saturated fatty acids, cholesterol and fructose, less severe APAP liver injury was observed despite enhanced hepatic CYP2E1 protein expression but CYP2E1 activity was not measured [56]. Adaptive responses in different antioxidant and anti-inflammatory pathways might explain this protective effect in this mouse model of obesity [56]. Conversely, higher APAP hepatotoxicity was observed in a mouse model of NASH despite unchanged CYP2E1 activity [59]. Although the reasons for the lack of CYP2E1 induction are unclear, it should be underlined that NASH was induced in this work with a methionine and choline-deficient (MCD) diet [59], which significantly reduces body weight and blood glycemia and does not cause systemic insulin resistance [9,60]. Hence, this peculiar metabolic profile might have removed some cues that otherwise might have led to CYP2E1 induction, as discussed in Section 4.2.2.

Hepatic CYP2E1 induction is a salient feature of obesity and NAFLD. Indeed, many clinical investigations consistently reported higher hepatic CYP2E1 expression and activity in patients with these metabolic diseases [9,43,80–85]. Hepatic CYP2E1 induction has also been found in many studies performed in different rodent models of obesity and NAFLD [49,57,82,86–91], although there are some exceptions as mentioned in Section 4.2.2.

Hepatic CYP2E1 induction in obese patients would not only cause more frequent or more severe APAP hepatotoxicity but may also favor the transition of fatty liver to NASH [82,89,92–94]. In steatotic hepatocytes, ROS overproduction secondary to CYP2E1 induction is indeed deemed to cause lipid peroxidation and the generation of noxious reactive aldehydes such as malondialdehyde (MDA) and 4-hydroxynonenal (4-HNE), which then promote necroinflammation and fibrosis [82,92–95]. Accordingly, the key role of CYP2E1 in NASH pathophysiology makes CYP2E1 inhibition or downregulation a promising therapeutic strategy in NAFLD [92,96,97].

The mechanisms of CYP2E1 induction in NAFLD are poorly understood. Accumulation of some fatty acids such as palmitic acid (C16:0) and stearic acid (C18:0) might play a role [74,98,99]. In keeping with the role of some fatty acids or lipids, a recent interventional study in healthy individuals showed that a short-term regular diet supplemented with whipped cream induced hepatic steatosis and significantly enhanced CYP2E1 activity [100]. Other mechanisms might involve hyperleptinemia, hyperglucagonemia and insulin resistance [10,25,82]. The exact downstream signaling pathways involved in CYP2E1 induction in NAFLD are still unknown.

### 4.1.2. Low Basal Levels of GSH

Low basal levels of liver GSH might also favor APAP hepatotoxicity in NAFLD as this is expected to hasten and even promote the profound GSH depletion taking place after APAP overdose (Figure 2A). Consequently, less NAPQI can be detoxified by hepatic GSH thus allowing the APAP reactive metabolite to covalently bind to different proteins and other cellular components, especially in mitochondria [16,17,23]. Significant reduction of basal levels of liver GSH has been reported in NAFLD, either in patients [101,102] or in some rodent models [103,104]. However, other animal investigations did not find any significant decrease in hepatic GSH content [49,50,105,106], although this was sometimes associated with higher levels of oxidized GSH (GSSG) [105].

The mechanisms that can cause low basal levels of liver GSH in NAFLD might be complex. Several factors might be involved including the extent of ROS overproduction via mitochondrial dysfunction, reduced synthesis of GSH and impairment of other antioxidant defenses, which can occur during the progression of NAFLD [37,105,107,108].

### 4.1.3. Extent of Steatosis and Accumulation of Deleterious Fatty Acids and Lipid Species

Investigations in genetically obese mice intoxicated with APAP showed that higher basal levels of hepatic triglycerides did not cause more severe APAP-induced hepatic cytolysis [49]. Hence, the extent of steatosis per se does not seem to favor APAP hepatotoxicity in NAFLD. In contrast, the accumulation of some fatty acids might specifically

favor liver injury. For instance, palmitic and stearic acids could be particularly harmful by promoting hepatic CYP2E1 induction [74,98,99], as previously mentioned. Furthermore, studies carried out in transgenic fat-1 mice, which endogenously convert n-6 PUFAs to n-3 PUFAs, showed that male animals were more susceptible to APAP-induced acute liver injury, possibly via a JNK-dependent mechanism and downregulation of signal transducer and activator of transcription 3 (STAT3) [109,110].

Cholesterol accumulation might also favor APAP hepatotoxicity in NAFLD (Figure 2A). Indeed, investigations in mice fed a high-cholesterol diet for 4 weeks showed more severe APAP-induced acute liver injury, possibly through the Toll-like receptor 9 (TLR9)/inflammasome pathway [111]. Interestingly, mitochondrial free cholesterol loading leads to mitochondrial GSH depletion in hepatocytes [112], which could promote mitochondrial dysfunction and cell death [113]. In contrast, CYP2E1 might not be involved because hepatic CYP2E1 expression and activity were reduced in rats fed a high-cholesterol diet for 11 weeks [114].

### 4.1.4. Mitochondrial Dysfunction

NAFLD is associated with complex mitochondrial alterations. In simple fatty liver, mitochondrial oxidative metabolism is stimulated, most probably as an adaptation to the increased levels of different substrates including fatty acids [37,38,108,115]. However, this adaptation can be lost in NASH, which is associated with reduced expression and activity of different mitochondrial respiratory complexes [37,38,108,115–117]. Accordingly, NASH-associated mitochondrial dysfunction might favor APAP hepatotoxicity (Figure 2A) since respiratory chain impairment is pivotal in APAP-induced liver injury [118–120]. However, there are currently no available data to confirm this hypothesis but different rodent models reproducing NAFLD progression can be useful for this [60,61,121].

### 4.1.5. Presence of Lobular Inflammation

Simple fatty liver can progress in some patients to NASH which is characterized by lobular inflammation, hepatocellular ballooning and the presence of some necrotic hepatocytes and apoptotic bodies, as previously mentioned. These pathological lesions are due at least in part to the overproduction of several proinflammatory cytokines such as tumor necrosis factor-α (TNF-α), interleukin-1β (IL-1β) and IL-6 [122,123], which could sensitize the liver to APAP-induced hepatotoxicity (Figure 2A). Interestingly, APAP-induced acute liver injury is exacerbated in Nlrp6$^{-/-}$ mice [124], a well-established mouse model of intestinal dysbiosis associated with enhanced gut–liver inflammatory responses [125]. However, it remains to be determined whether obesity- and NAFLD-associated gut dysbiosis [126,127] could play a role in higher APAP hepatotoxicity in these metabolic diseases. Other investigations showed that hepatic inflammation favors liver injury induced by different drugs and chemicals [128–130].

### 4.2. Factors That Could Mitigate APAP Hepatotoxicity in Obesity and NAFLD
### 4.2.1. Alteration in APAP Absorption and Distribution

Only a few clinical studies dealt with the impact of obesity on gastrointestinal absorption of APAP and its whole-body distribution. To our knowledge, only one study reported a lower absorption rate of APAP in obese subjects, which was associated with a decrease in the maximum plasma concentrations of the pain reliever [131]. Regarding whole-body distribution, two studies reported higher APAP volume of distribution (Vd) in obese subjects [43,132]. However, all these investigations were carried out in morbidly obese persons and further investigations would be needed to confirm these data for body mass index (BMI) below 40 kg/m$^2$. Nevertheless, decreased APAP gastrointestinal absorption and higher Vd could favor lower APAP plasma and liver concentrations, at least in some obese patients [9].

### 4.2.2. Lack of CYP2E1 Induction or CYP2E1 Downregulation

Although hepatic CYP2E1 activity is frequently increased in NAFLD (see Section 4.1.1), some investigations reported a lack of CYP2E1 induction, which might not allow NAPQI overproduction (Figure 2B). Indeed, several clinical studies showed that some obese patients had CYP2E1 activity in the range of nonobese individuals [43,46,85,133]. Experimentally, hepatic CYP2E1 expression and activity are not increased in obese leptin-deficient ob/ob mice and leptin receptor-deficient fa/fa Zucker rats [49,68,134]. Although this might suggest that the leptin signaling pathway is needed for CYP2E1 induction in obesity and NAFLD, hepatic CYP2E1 activity is enhanced in leptin receptor-deficient db/db mice, especially in females [49]. The very high glycemia and ketonemia in these mice [49,66] might play a role in CYP2E1 induction in this context of severe diabetes [10]. Because many endogenous molecules, hormones and cytokines are deemed to regulate hepatic CYP2E1 expression and activity, sometimes with opposite effects [10,82,135–137], it is possible that CYP2E1 induction might not always occur in obesity and NAFLD.

Another possibility could be the loss of CYP2E1 induction during NAFLD progression (Figure 2B). Indeed, recent investigations suggested that CYP2E1 induction seems to wane when NASH progresses toward advanced fibrosis [57], in line with clinical data reporting a significant reduction of CYP2E1 expression with the progression of liver fibrosis [138,139]. Increased production of proinflammatory cytokines including TNF-α might play a role in this progressive decline of CYP2E1 expression [135,139]. In contrast, the profibrotic cytokine transforming growth factor-beta (TGF-β) does not seem to be involved in fibrosis-associated CYP2E1 downregulation [140,141].

### 4.2.3. Reduced CYP3A4 and CYP1A2 Activity

CYP3A4 (also referred to as CYP3A) and CYP1A2 are also involved in APAP biotransformation to NAPQI, although to a lesser extent than CYP2E1 [26,27]. Many clinical and experimental studies consistently reported lower hepatic expression and activity of CYP3A4 in obesity and NAFLD [83,142–150]. Hence, lower CYP3A4 activity in obesity and NAFLD might reduce the generation of NAPQI after an APAP overdose (Figure 2B).

Several clinical studies on CYP1A2 activity reported little or no change in obesity [83,147,149]. Interestingly, investigations in patients with NAFLD reported that CYP1A2 expression and activity were unaltered in fatty liver but significantly reduced in NASH [151,152]. These data seem to be in line with the investigations carried out in obese patients since NASH occurs only in a minority of those people [35]. In rodent models of NAFLD, CYP1A2 expression and activity were significantly decreased in most investigations [153–158], but increased or unchanged in some others [143,159]. Like CYP3A4, lower CYP1A2 activity in NAFLD might also reduce the generation of NAPQI after an APAP overdose (Figure 2B).

### 4.2.4. Increased APAP Glucuronidation

Clinical and experimental investigations consistently reported increased APAP glucuronidation in obesity and NAFLD [9,43,49,83,160,161], which is expected to reduce the extent of APAP bioactivation to NAPQI (Figure 2B). Of note, UGT1A6 and UGT1A9 are the main UGT isoforms involved in APAP glucuronidation in humans [162] but only UGT1A9 protein expression tended to be increased in patients with obesity-related fatty liver [163].

### 4.2.5. Exposure and Accumulation of Protective Fatty Acids

Two studies carried out in female transgenic fat-1 mice (which endogenously convert n-6 PUFAs to n-3 PUFAs) showed significant protection against APAP-induced acute liver injury [109,164]. In the study by Liu et al., the opposite effect was observed in male mice and this gender difference was attributed to estrogens [109]. Of note, the expression of hepatic CYP2E1 in female mice was unchanged in one study [164], whereas CYP2E1 was not investigated in the second one [109]. Other investigations in rats showed that dietary supplementation with the n-3 polyunsaturated eicosapentaenoic and docosahex-

aenoic acids (EPA and DHA) protected against acute APAP liver injury [165]. According to the authors, the hepatoprotective effect of n-3 PUFAs against APAP liver injury might be mediated via their anti-inflammatory and antioxidant properties [164,165]. Another study in rats fed a diet with 20% fish oil (i.e., rich in n-3 PUFAs) also reported protection against APAP-induced acute liver injury, which was deemed to be related to higher APAP glucuronidation [166]. Interestingly, n-3 PUFAs reduced hepatic CYP2E1 activity in insulinopenic diabetic rats [167] but their protective effect against APAP hepatotoxicity was not investigated in this study.

## 5. APAP-Induced Liver Injury after Bariatric Surgery

Roux-en-Y gastric bypass and sleeve gastrectomy are surgical procedures increasingly used for the treatment of morbid obesity and comorbidities including NAFLD [168,169]. A retrospective study suggested that weight loss surgery may predispose to acute liver failure after APAP overdose and this was independent of alcohol abuse and the use of APAP–narcotic combination drugs [170]. More recently, a case of fulminant hepatitis was observed after laparoscopic sleeve gastrectomy in a young woman who received therapeutic doses of APAP [171]. In addition to malnutrition and vitamin deficiency, the authors pointed to other possible risk factors including rapid weight loss, which might have aggravated preexisting fatty liver [171]. Notably, although CYP2E1 activity in obese patients decreases after bariatric surgery it remains higher than in healthy volunteers [133,172]. Thus, increased CYP2E1 activity might favor APAP-induced liver injury in obese patients even after such surgery. However, beyond CYP2E1 activity, other metabolic parameters most probably explain the profound alterations of APAP bioavailability observed after weight loss surgery [173,174]. Hence, further investigations would be needed to determine the mechanisms whereby bariatric surgery might predispose to APAP hepatotoxicity.

## 6. APAP-Hepatotoxicity in Type 1 Diabetes Mellitus

Type 1 diabetes mellitus (T1DM) is a chronic autoimmune disease caused by insulin deficiency and leading to severe hyperglycemia [175]. Importantly, the pathogenesis of T1DM significantly differs from that of obesity-related T2DM [176,177]. Nonetheless, T1DM seems to be frequently associated with fatty liver, which can progress to steatohepatitis and cirrhosis in some patients [178,179]. Some clinical investigations disclosed that diabetes increases the risk and the severity of DILI but these studies did not specify whether there was a difference between T1DM and T2DM [180–182]. Moreover, these investigations did not provide specific information on APAP.

T1DM can be induced in rats and mice by single or repeated injections of streptozotocin, a pancreatic β-cell poison [10,183]. Using this experimental model, a recent study reported that APAP-induced acute liver injury was exacerbated in diabetic mice possibly via a hyperglycemia-induced proinflammatory response in liver Kupffer cells [184]. Although not investigated in this study, it is possible that CYP2E1 induction might also have played a role in liver injury aggravation [10]. Indeed, numerous studies (but not all—see below) showed that streptozotocin-induced diabetes is associated with higher hepatic CYP2E1 protein expression and activity [10,167,185–188].

Contrasting with the study by Wang et al. [184], several investigations in streptozotocin-treated rodents showed that T1DM protected against APAP-induced acute hepatotoxicity [189–191]. The exact reasons for these discrepancies are unknown although higher APAP glucuronidation and improved liver repair in diabetic animals might play a role [189–191]. However, it is worth mentioning that hepatic CYP2E1 activity was not increased in these studies, thus contrasting with many other investigations reporting CYP2E1 induction in streptozotocin-treated rodents [10,167,185–188]. Further studies would be needed in order to determine why hepatic CYP2E1 induction is not always observed in streptozotocin-induced experimental diabetes. The extent of insulinopenia, ketonemia and hyperglycemia might be pivotal [10], in addition to other metabolic factors already discussed in this review.

## 7. Conclusions

Although obesity and NAFLD appear to increase the risk or the severity of APAP-induced acute liver injury, this relationship has not always been reported. As discussed in previous reviews [9,13,46] and this one, we propose that the occurrence and outcome of APAP-induced liver injury in these metabolic diseases might depend on a subtle balance between metabolic factors that can be protective for the liver and others that favor the generation of NAPQI (Figure 2). Hence, further investigations are needed in order to understand why some obese individuals could be at risk for APAP-induced hepatotoxicity and why some others are not. Although the absence of hepatic CYP2E1 induction might explain the lack of increased risk, other mechanisms might be involved including reduced APAP gastrointestinal absorption, enhanced Vd, higher hepatic glucuronidation and lower hepatic CYP3A4 activity (Figure 2B). In contrast, robust CYP2E1 induction, lobular inflammation, low basal concentrations of hepatic GSH and NASH-associated mitochondrial dysfunction might favor APAP hepatotoxicity in obesity and NAFLD (Figure 2A). While some of these factors are difficult to investigate in patients, many rodent models can be useful for mechanistic purposes [60,61,121]. Of note, these rodent models of obesity and NAFLD could also be valuable in order to determine whether repeated or chronic administration of APAP at therapeutic doses can cause more severe liver injury.

From a clinical viewpoint, physicians are encouraged to carry out regular monitoring of liver function in obese patients treated with chronic APAP administration, in particular in patients with pre-existing NAFLD. Finally, it should be underlined that chronic ethanol consumption constantly causes hepatic CYP2E1 induction while recent investigations reported that alcohol consumption and obesity (or metabolic syndrome) can synergistically augment the risk and severity of steatohepatitis, cirrhosis and HCC [192–194]. Hence, further investigations would be required to determine whether obese people who regularly consume alcohol have an even higher risk of APAP-induced hepatotoxicity.

Finally, a major issue for the future is to better prevent liver failure and mortality after APAP overdose, irrespective of the patient's body weight. Although NAC is the only approved antidote to treat APAP-induced liver injury [10,14], other therapeutic compounds are currently being developed to inhibit CYP2E1 activity (fomepizole), or to prevent mitochondrial oxidative stress (MitoTEMPO) and peroxynitrite formation (calmangafodipir) [195]. Numerous phytochemicals with efficient antioxidant properties might also be promising antidotes [196]. As for NAC, these compounds might be able to protect against APAP-induced necrosis [17,197] and other possible types of cell death including necroptosis and apoptosis [198,199]. Furthermore, targeting autophagy, mitophagy and mitochondrial biogenesis could also be promising therapeutic strategies [22,195,200].

**Author Contributions:** Conceptualization, K.B., J.M. and B.F.; writing—original draft preparation, B.F.; writing—review and editing, K.B., C.P., P.B.-G. and J.M.; visualization, K.B., J.M. and B.F.; supervision, B.F.; project administration, B.F. All authors have read and agreed to the published version of the manuscript.

**Funding:** This research received no external funding.

**Institutional Review Board Statement:** Not applicable.

**Informed Consent Statement:** Not applicable.

**Data Availability Statement:** Not applicable.

**Acknowledgments:** We are grateful to the Institut National de la Santé et de la Recherche Médicale (INSERM) for its constant support.

**Conflicts of Interest:** The authors declare no conflict of interest.

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
