# Peer review of "Acetaminophen-Induced Hepatotoxicity in Obesity and Nonalcoholic Fatty Liver Disease: A Critical Review"

_livers, doi:10.3390/livers3010003_

Round 1
Reviewer 1 Report
The review is quite well written. I'd like to compliment the authors on organizing all the pertinent material so well.
It is currently unclear if NAFLD increases a person's risk of developing acute liver failure brought on by APAP overdose. Clinical and experimental research has been done on the subject, but no one has yet developed a singular viewpoint. If the authors could include a table identifying contradictory studies and their findings, it would significantly enhance the value of this review.
Author Response
We would like to warmly thank Reviewer 1 for his (her) compliments and remarks regarding the potential interest to include a table. Accordingly, we included in the revised manuscript two tables, respectively summarizing clinical studies and rodent investigations on APAP-induced acute liver injury in obesity and NAFLD.
Reviewer 2 Report
This is a very necessary review that discusses the occurrence and outcome of APAP-induced liver injury in obese individuals with NAFLD and how it is dependent metabololic factors that increase or decrease hepatotoxicity. I only have very minor comments:
-Please stardardize Abbreviations (sometimes APAP, others acetaminophen, HCC, hepatocellular carcinoma, Vd...)
-I would replace pain killer by pain reliever or analgesic/antipyretic
-Important mechanisms in APAP-induced hepatotoxicity in NAFLD is are necrosis, apoptosis, necroptosis and autophagy which should be at least mentioned.
-Is treatment with antioxidants an plausible option for these patients?
Author Response
We would like to thank Reviewer 2 for his (her) nice comments on our review. Regarding the minor remarks, we standardized in the revised version the abbreviations and also replaced the term “painkiller” by “pain reliever” or “analgesic and antipyretic drug”. We also added in the conclusion a paragraph pertaining to the therapeutic compounds that are being developed to treat APAP-induced liver injury, including antioxidants and CYP2E1 inhibitors. We also added two sentences to mention other types of cell death and to underline that, in addition to antioxidants, targeting autophagy, mitophagy and mitochondrial biogenesis could also be promising therapeutic strategies. This paragraph is as follows: “Finally, a major issue for the future is to better prevent liver failure and mortality after APAP overdose, irrespective of the patient’s body weight. Although NAC is the only approved antidote to treat APAP-induced liver injury [10,14], other therapeutic compounds are currently being developed to inhibit CYP2E1 activity (fomepizole), or to prevent mitochondrial oxidative stress (MitoTEMPO) and peroxynitrite formation (calmangafodipir) [195]. Numerous phytochemicals with efficient antioxidant properties might also be promising antidotes [196]. As for NAC, these compounds might be able to protect against APAP-induced necrosis [197] and other possible types of cell death including necroptosis and apoptosis [198,199]. Furthermore, targeting autophagy, mitophagy and mitochondrial biogenesis could also be promising therapeutic strategies [22,195,200].”
Reviewer 3 Report
This review summarized the findings of acetaminophen-induced hepatotoxicity in obesity and nonalcoholic fatty liver disease in both human and rodent studies and their potential mechanisms. NAFLD and APAP toxicity are two prevalent liver diseases, thus this topic is interesting and important to the field. The review is also well-written and significant to both liver field and communities.
Author Response
We would like to heartfully thank Reviewer 3 for his (her) compliments and remarks regarding the potential interest and significance of our review.